# Effect of Graphene Oxide Coating on Natural Fiber Composite for Multilayered Ballistic Armor

**DOI:** 10.3390/polym11081356

**Published:** 2019-08-16

**Authors:** Ulisses Oliveira Costa, Lucio Fabio Cassiano Nascimento, Julianna Magalhães Garcia, Sergio Neves Monteiro, Fernanda Santos da Luz, Wagner Anacleto Pinheiro, Fabio da Costa Garcia Filho

**Affiliations:** Department of Materials Science, Military Institute of Engineering-IME, Rio de Janeiro 22290270, Brazil

**Keywords:** curaua fibers, graphene oxide coating, epoxy composites, ballistic performance

## Abstract

Composites with sustainable natural fibers are currently experiencing remarkably diversified applications, including in engineering industries, owing to their lower cost and density as well as ease in processing. Among the natural fibers, the fiber extracted from the leaves of the Amazonian curaua plant (*Ananas erectifolius*) is a promising strong candidate to replace synthetic fibers, such as aramid (Kevlar™), in multilayered armor system (MAS) intended for ballistic protection against level III high velocity ammunition. Another remarkable material, the graphene oxide is attracting considerable attention for its properties, especially as coating to improve the interfacial adhesion in polymer composites. Thus, the present work investigates the performance of graphene oxide coated curaua fiber (GOCF) reinforced epoxy composite, as a front ceramic MAS second layer in ballistic test against level III 7.62 mm ammunition. Not only GOCF composite with 30 vol% fibers attended the standard ballistic requirement with 27.4 ± 0.3 mm of indentation comparable performance to Kevlar™ 24 ± 7 mm with same thickness, but also remained intact, which was not the case of non-coated curaua fiber similar composite. Mechanisms of ceramic fragments capture, curaua fibrils separation, curaua fiber pullout, composite delamination, curaua fiber braking, and epoxy matrix rupture were for the first time discussed as a favorable combination in a MAS second layer to effectively dissipate the projectile impact energy.

## 1. Introduction

In recent decades, the increasing efficiency of ballistic armors has emerged as a relevant factor in personal and vehicular security, for both civilian and military protection. The search for lighter and stronger armor materials has been increasing in proportion to the escalating power and sophistication in firearms development [1]. Research works are showing that polymer composites reinforced with natural lignocellulosic fibers (NLFs) present ballistic efficiency in multilayered armor systems (MAS), with front ceramic, comparable to synthetic aramid fabric, such as Kevlar™ [1,2,3,4,5,6,7,8,9,10,11,12,13,14,15,16,17]. In general, NLF composites have the advantage of environmental sustainability in association with cost-effectiveness, lower density, and easy fabrication as compared to synthetic fibers composites [18,19,20,21].

Together with ballistic protection, recent works on nano and micro cellulose [22,23,24,25,26,27], are also disclosing special applications for NLFs. Among the several papers on ballistic application of NFL composites for MAS second layer stands those using curaua fibers (CF) [1,6,7,10,11,17]. This fiber, native of the Amazonian region, is extracted from the leaves of a plant, *Ananas erectifolius*, sharing the pineapple family. It has attracted considerable interest as polymer composite reinforcement [28,29,30,31,32,33,34] owing to relatively lower density (0.96 g/cm^3^) in comparison to glass (2.58 g/cm^3^) and aramid (1.44 g/cm^3^) synthetic fibers [35]. In consequence, the CF specific tensile strength (~2.2·GPa.cm^3^/g) is higher than that of glass (~1.4 Gpa·cm^3^/g) and close to that of aramid (~2.8·Gpa.cm^3^/g) fibers.

As most NLFs applied in polymer composite [36,37,38,39,40,41], the curaua fiber also displays low interfacial shear strength, associated with poor fiber adhesion, while reinforcing a polymer matrix. This is due to their amorphous hemicellulose and lignin that act as natural hydrophilic wax adsorbing water on the fiber surface. Consequently, a weak bonding is expected to exist between the surface of the curaua fiber and the hydrophobic polymer such as polyester [29] and epoxy [37]. This affects the composite performance as MAS second layer for ballistic protection. Indeed, the impact of a high velocity projectile against a MAS with curaua composite results in different fracture mechanisms including delamination and matrix cracking pattern as well as fiber rupture and pullout [7,10,11]. Some of these mechanisms are essential for impact energy. However, others like delamination can impair the integrity of the composite target after a first ballistic shooting. This causes loss of its ability to protect against serial shootings as required by the standard [42].

In spite of the comparable ballistic performance to a same thickness Kevlar™ laminate as MAS second layer, the integrity of a NLF composite is always questionable. Lower amounts, usually less than 30 vol%, of fiber were found to result in composite shattering [4,5,8,9,11,12,13,14,15,16,17]. Even a 30 vol% NLF composite may be split by delamination, i.e., decohesion between fiber and matrix, which allows easy perforation of the projectile in case of a second shooting. Surface modification of NLFs has extensively been applied to improve the fiber matrix adherence [43,44]. This will be an effective way to prevent delamination.

Since the rise of graphene [45], it has increasingly been studied and investigated for possible technological applications. In particular, graphene has attracted a considerable attention for its superior performance as composite reinforcement owing to outstanding mechanical properties [46]. The direct oxidation of graphite is considered as an alternative route for producing substantial quantities of another remarkable material, the graphene oxide (GO). Studies conducted on the properties of GO revealed good chemical reactivity and easy handling owing to its intrinsic functional groups in association with amphiphilic behavior [47,48]. Among the several methods reported, to improve NLF composite adhesion and prevent lamination, only few have today been dedicated to graphene or graphene oxide coating [45,49,50].

To the knowledge of the authors of the present work, GO has not yet been applied as a coating onto NLFs to improve interfacial shear strength with respect to a polymer composite for armor application. More specifically, as a novel method to provide efficient fiber/matrix interface for impact energy dissipation. Therefore, the objective of this work is, for the first time, to investigate the ballistic performance of 30 vol% graphene oxide coated curaua fiber (GOCF) reinforced epoxy composite, as a MAS second layer against the treat of level III [42] high velocity projectile. In addition to the comparison of GOCF with both non-coated 30 vol% CF epoxy composite and same thickness Kevlar™, this work also investigates the integrity condition of these composites.

## 2. Materials and Methods

Curaua fibers, shown in Figure 1a, were supplied by the University of Pará (UFPA), Belém, Brazil. The polymer used as matrix was a commercially available epoxy resin, bisphenol A diglycidyl ether type (DGEBA), hardened with triethylene tetramine (TETA), using the stoichiometric ratio of 13 parts of hardener per 100 parts of resin, fabricated by Dow Chemical, São Paulo Brazil, and distributed by Resinpoxy Ltda (Rio de Janeiro, Rio de Janeiro).

Curaua fibers were used in two main conditions, namely: as-received, non-coated fibers (CF), and graphene oxide coated fibers (GOCF). Initially the as-received fibers were subjected to a mechanical treatment using a hard bristle brush for cleaning, separation, and fiber alignment. Then fibers were cut into 150 mm in length and placed in an oven at 80 °C for 24 h until the fiber weight remained stable. This corresponds to the as-received CF used to produce plain composite plates.

The GO used in this work was produced by the Hummers Offeman method, modified by Rourke et al. [47]. The CFs, that have already passed the brush and drying stages, were then immersed in a 0.56 mg/mL GO solution corresponding to 0.1% of weight of the fiber and kept under agitation for 30 min in a universal mechanical shaker, in order to guarantee and optimize the contact of the GO with the fiber. Thereafter the CF soaked with GO dispersion were placed in an oven at 80 °C for 24 h, obtaining at the end the GOCF. Raman spectroscopy analysis was conducted in a model NTEGRA Spectra equipment to certify the existence of GO layers on the fiber surfaces.

To fabricate the composite plates, a metal mold with dimensions of 150 × 120 × 12 mm was used. The plates were processed in a SKAY hydraulic press by applying a load of 5 tons for 24 h. For the CF, the density of 0.92 g/cm^3^ [31] was used as the initial reference and 1.11 g/cm^3^ for the epoxy resin [35]. The percentages of both CF and GOCF studied in this work was 30 vol%. Figure 1 shows the general macroscopic aspect of (a) CF, (b) GOCF, (c) and their corresponding epoxy composites.

Interfacial shear strength tests were performed to investigate the influence of GO coating onto curaua fiber in curaua-epoxy composites. For this, the method described by Kelly and Tyson [51] was used. The measured parameters were the critical length and the interfacial shear strength. Tensile tests of the individual fiber were carried out according to ASTM D 3822-01 standard [52]. The test used a support (frame) made of paper and plaster, in order to keep the fiber stretched and firmly attached to facilitate the positioning in the grips of the model 3365 Instron equipment. A 25 KN load cell and a strain rate of 5 mm/min were used to perform each individual fiber specimen tensile test. Ten specimens for each test condition were used for both CF and GOCF, with a gage length of 40 mm. The fiber diameter was measured by an optical microscope Olympus BX53M. Before starting the test, the paper is cut to avoid interference in the tensile results.

Ballistic tests were carried out to investigate both CF and GOFC composites capacity of dissipating kinetic energy of a high velocity projectile in a MAS. The MAS used in this work consist, of a front layer of ceramic, an intermediate layer made from both the CF and GOCF epoxy composites, Figure 2. The MAS is placed over a 50 mm thick clay witness (CORFIX™), which has a similar consistency as a human body. The ballistic test system is illustrated in Figure 3. The objective is to obtain the measurement of the trauma, also known as backface signature (indentation) caused by the impact of the 7.62 mm caliber ammunition on the MAS target. According to the NIJ 0101.04 standard [42] a ballistic armor will be effective if the indentation caused in the clay witness is equal to or less than 44 mm. Measurements were performed with a Q4X Banner digital laser sensor. The tests were carried out at the Brazilian Army Assessment Center (CAEx), Rio de Janeiro.

Microscopic analyses of the curaua fibers and fractured surface of the investigated composites were performed by scanning electron microscopy (SEM) in a model Quanta FEG 250 Fei microscope operating with secondary electrons between 5 and 10 KV. The energy dispersive spectroscopy (EDS) analyses were performed using a Bruker Nano GmbH XFlash 630M detector.

The FTIR technique was used to investigate the possible influences of GO on the functional groups of the curaua fibers, in an IR-Prestige-21 model spectrometer from Shimadzu, using the transmittance method with the KBr insert technique. For all samples, the same mass quantities of 2 mg of fiber and 110 mg of KBr were used.

For the analysis by thermogravimetry (TGA), the curauá fibers in CF, GOCF, and its composites were comminuted and placed in aluminum crucible of the TA Instruments, model Q 500 analyzer. Samples were subjected to a heating rate of 10°/min, starting at 30 °C up to 700 °C.

The thickness estimation of GO coating was obtained by atomic force microscope in a model Park systems XE7 atomic Force Microscope.

## 3. Results and Discussions

The Raman spectra of GO is shown in Figure 4. The intensity ratio of the D and G bands (ID/IG) revealed structural defects and the indication of disorder. The (ID/IG) ratio was calculated as 1.032:1, in accordance with previous authors [47,48]. Besides, a broad and shifted to higher wavenumber of 2D band was seen at 2720 cm^−1^ for GO in Figure 4. 2D band can be used to determine the layers of graphene (monolayer, double layer or multilayer) as it is highly sensitive to stacking of graphene layers. Thus, the location of 2D band confirms that the produced GO was multilayer. A monolayer graphene is normally observed at 2679 cm^−1^ from the spectra. In addition, the shifted location of 2D band, because of the presence of oxygen-containing functional groups, prevents the graphene layer to stack [49].

The main absorption bands of the CF fiber spectrum can be seen as: 3379 cm^−1^, which is related to the elongation of OH groups present in cellulose and water. The 2916 cm^−1^ band can be attributed to the symmetrical and asymmetrical stretching (C–H) of the aliphatic chain, 1736 cm^−1^ corresponding to the acid elongation vibration (C=O); 1430 cm^−1^ (aliphatic C–H vibration) and 1110 cm^−1^ from the elongation vibration of the ether groups. Other bands refer to the existence of high content of oxygen functional groups on GOCF surface, such as (–C–O–C) and (–C–OOH) [53]. Chemical treatments or modifications of major fiber surface groups (–OH) can be very valuable in detecting and confirming the type of new bond established on the fiber surface and the interaction with the polymer in the case of fiber reinforced polymers [31].

With GO coating, even at low concentrations, several changes in the spectra can be seen in Figure 5. The relative intensities between some bands have changed, suggesting that the GO molecule may have linked to the functional groups such as those mentioned above, reducing almost all intensities of the spectrum. In addition, the absorption band at 1649 cm^−1^ may refer to vibrations of the present GO skeletal ring [54].

The light band that can be seen at 1560 cm^−1^ can be attributed to the vibrations of benzene rings present in GO [55]. In addition, with the cure of GO curaua fibers, the absorption bands at 833 cm^−1^ (C–H out of plane for *p*-hydroxyphenyl units) reduced the intensity [56], suggesting that the GO caused changes in the CF fiber functional groups, such as the hydroxyl and carboxyl groups of GO sheets react with the hydroxyl groups of CF, resulting in better wettability between CF and epoxy matrix [49].

The onset of the degradation step was observed at approximately 64 to 150 °C in both CF and GOCF. This effect may indicate the evaporation of moisture absorbed by the fibers. The main mass degradation step was observed starting at 293 °C for CF and 300 °C for GOCF fibers as can be seen in Figure 6. According to some studies, this indicates the stages of hemicellulose, cellulose, and lignin degradation, respectively [33,57,58,59]. The residue generated by CF fibers was 15% and by GOCF it was 14%.

In the differential thermal analysis (DTA) curves as shown in Figure 7, for the CF fibers, three stages were observed: the first one was between 250 and 300 °C, referring to the decomposition of the hemicellulose, with maximum degradation rate at 272 °C. The second process occurred between 293 and 350 °C, with a maximum degradation rate around 327 °C, which may be related to decomposition of cellulose. Lignin decomposition occurred in the third stage, between 400 and 450 °C, with a maximum degradation rate of around 422 °C. However, a distinct behavior was presented by the GOCF fibers. Their degradation was shifted to higher temperatures and this effect may indicate an increase in the thermal stability of the fibers [33,57,58,59].

The degradation ratio of the different fibers, CF and GOCF, in Figure 6 and Figure 7 indicate that by the presence of graphene oxide increases the thermal stability of the temperature at 7 °C, with the onset around 300 °C. The effect may be due to the formation of an insulation to heat propagation by the GO coating which retarded degradation and improved thermal stability [50].

Figure 8 shows SEM images of both curaua fibers investigated: (a) as-received non-coated (CF) and (b) graphene oxide coated (GOCF). Average diameter measurements conducted in 10 fibers for each case revealed values of 54.2 ± 14.3 μm for the CF and 51.1 ± 12.0 μm for the GOCF. As expected, these values are practically the same within the standard deviation. This indicates that the graphene oxide coating did not affect the fiber diameter. In fact, the GO coating was estimated to be approximately 10 nm. This would correspond to a negligible increase of less than 10^−3^ vol% in the composite volume fraction of curaua fiber. Since the thinner GO coating cannot affect the curaua fiber strength.

With higher magnification, Figure 9 illustrates the different surface aspects of CF and GOCF. One should notice the uniform smoother surface of the GOFC, Figure 9b, because of the graphene oxide coating as compared with the rougher CF surface in Figure 9a. The GO sheets, prepared by the modified Hummers method, form a stable and homogeneous suspension and exhibit a typical transparent wavy aspect, when coated on the fibers as shown in Figure 9b.

With higher magnification, one notes that the CF fiber is not very stable under the electron beam, with cracks opening on its surface as can be seen in the Figure 9a and indicated by a white arrow. On the other hand, GOCF fiber is more thermally stable, not reacting with the heat generated by the electron beam during the image acquisition, which corroborates the TGA results.

Through the EDS analysis in Figure 10 it was possible to identify the elements present on the surface of both CF and GOCF fibers. For the CF, only carbon and oxygen were identified, the other peaks of the spectrum refer to the copper used in the covering of the fibers. For GOCF fibers, besides carbon and oxygen, it was identified phosphorus and sulfur, which are residues of the reagents used for the production of GO. It can be noted that for CF fibers the C/O ratio is 0.91 decreasing to 0.14 for GOCF fibers, due to the presence of oxygen-containing functional groups in GO [50].

Table 1 presents tensile test results for both curaua fibers, CF and GOCF, with regard to the ultimate stress $\left(\sigma_{f}\right)$, total strain (ε), and Young’s modulus (**E**). Regarding this table, it is important to mention that the values obtained for these properties agree with those reported in the literature [29,31].

One may infer from the results in Table 1 that the GO coating caused an increase of 47.8% in the Young’s modulus of the fibers, corroborating to other authors [49,50]. On the other hand, in maximum tension there was a 71.9% reduction showing a more brittle but more rigid behavior of GOCF compared to CF, possibly because of the relatively low amount of GO used, forming a very thin film on the surface. In addition, it will be shown that, as reinforcement of epoxy matrix composite, this coating is associated with relevant differences in terms of fiber/matrix adherence.

Figure 11 shows the pullout curves, based on the Kelly and Tyson method [51], the curve has three levels, corresponding to the failure mechanisms that occurs in the composite, the first one, for short embedded lengths, refers to the level where only the fiber pullout occurs. In the second stage, there are pullout and fiber rupture; at this stage the length of the fiber in the composite has already reached but not exceeded the critical length. However, when the critical length is exceeded, as the case of the third stage, the failure mechanism of the composite is only by rupture of the fibers, i.e., there are no longer the pullout mechanism. Thus, the critical length for the system fiber/matrix is defined by the maximum value associated with the first stage of the curve [30]. The value of the fiber critical length was calculated as $l_{c}$ = 2 mm for the CF/epoxy $l_{c}$ = 1 mm for the GOCF/epoxy. These values are much lower than that of $l_{c}$ = 10.2 mm, reported for curaua fiber/polyester [30]. The GOCF/epoxy critical length is sensibly lower than non-coated CF/epoxy. Consequently, the interfacial shear strength of the GOCF/epoxy, $\tau_{c}$ = 27.5 MPa is more than 50% higher than that of the CF/epoxy, $\tau_{c}$ = 18.2 MPa. One may infer that for the same embedded length the CF fiber pullout voltage is greater than GOCF fiber, however, this behavior is due to the fact that the GOCF composite is now a new system with a new fiber/matrix interface [30]. Therefore, for each system, there is a strength and a certain critical length. As the fibers had their tensile strength affected by the GO coating, it is expected that the fiber/matrix system strength also presents similar behavior.

In the present work, for the first time, ballistic tests were carried out to measure the trauma on the witness clay in MAS target with a second layer epoxy matrix composite reinforced with 30 vol% of curaua fiber both CF and GOCF. In none of the MAS tested, there was complete perforation of the 7.62 mm projectile and the indentation of the clay witness was less than 44 mm, a value considered to be non-lethal to humans by the standard [42]. The results obtained are presented in the Table 2 and visualized in Figure 8. They were also compared with other MAS using distinct fibers, as well as with a same thickness laminate of Kevlar™, as a second layer. The limit value established by the standard is shown as an upper dashed horizontal line, in Figure 12. These results were found to be in good agreement with other authors [4,15] and relatively better than those by Braga et al. [7].

In Figure 12, one should note a slight increase in the value of the indentation in the clay witness caused by the 7.62 mm projectile impact against a MAS target with GOCF epoxy composite as a second layer. Figure 13 illustrates the aspect of both MASs, with CF and GOCF composites, before and after the ballistic test. The integrity, an essential factor for practical applications, is shown to be better than the GOCF in comparison to MAS with CF epoxy composite as a second layer. Indeed, in this latter, the plate fractured into two large pieces as can be seen in Figure 13b. By contrast, MAS target with GOCF composite remained relatively intact in Figure 13d.

The smaller hexagonal ceramic tiles, front MAS layer in Figure 13a,c are completely destroyed, Figure 13b,d upon the projectile impact. In an actual armor vest, these tiles compose a mosaic to allow multiple shootings in which a single tile is hit at a time without compromising the armor protection. Figure 14 shows by SEM the ruptured surface of a tile ceramic totally destroyed. This rupture occurs by intergranular fracture absorbing most of the kinetic energy of the projectile. The magnified image in Figure 14b, displays in detail an intergranular microcrack associated with this mechanism of fracture, similar to what was verified by other authors [15,34].

Another important participation of the composite plate as MAS second layer is the capture of ceramic fragments resulting from the shattered front ceramic, Figure 14, which corresponds to a significant amount of the absorbed impact energy [60]. Figure 15 illustrates the capture of ceramic fragments by curaua fibrils that compose each curaua fiber in the epoxy composite. In this figure it is important to note not only the extensive incrustation of microfragments covering the fibrils but also effective fibrils separation. Indeed, as shown in Figure 15 and Figure 16 like most LNFs a curaua fiber is composed of well-adhered fibrils that split apart when subjected to an applied stress [29]. The shock wave resulting from the projectile impact in the present ballistic tests, Figure 3, in addition to complete shatter the front ceramic, Figure 14, also caused separation of fibrils clearly shown in Figure 15. Therefore, for the first time, it is reported a whole view of the mechanisms responsible for dissipating the remaining energy, after the projectile impact against the front ceramic, by the curaua fiber composite as MAS second layer. The indentation results in Table 2, indicate that these mechanisms are responsible for a ballistic performance comparable to Kevlar™ laminate, which is a much stronger material. While the Kevlar™ mechanisms of energy absorption, as MAS second layer, is basically the capture of fragments [60], the curaua fiber composite is associated with several mechanisms with distinct participation of the GO coating. The combination of the following mechanisms makes both CF and GOCF epoxy composites in Table 2 as effective as Kevlar™.

Capture of fragments, Figure 15, the same mechanism first shown in Kevlar™ [26] and later reported for curaua fiber [11,17] and non-woven curaua fabric [7] polymer composites. Apparently, this capture of fragments is not affected by the GO coating.

Fibrils separation, also illustrated in Figure 16, is a specific mechanism for stress-subjected curaua fibers [29], which contributes to dissipate energy by generating free surface area between fibrils. Observed evidences suggest that GO coating makes difficult the fibril separation and has, comparatively, a reduced dissipated energy. This separation in plain curaua fibers (CFs) might disclose individual nano and micro cellulose chains with special behavior [22,23,24,25,26,27].

Fiber pullout shown in Figure 17 in which a hole left in one site of the fracture surface was caused by a curaua fiber pullout. The insert with higher magnification revels a remaining attached fibril separated from the pulled fiber. In this case, energy is dissipated by the created hole/pulled-out fiber-free surface. No evidence of pullout was found in the GOCF composites, which also indicates a reduced impact energy absorption.

Composite delamination, Figure 13b, which is a macro mechanism of energy dissipation involving the creation of relatively large free surface area associated with the extensive separation between curaua fiber/epoxy matrix. As aforementioned, delamination impairs the integrity of the 30 vol% CF composites despite the dissipated impact energy. In contrast, delamination is not effective in the 30 vol% GOCF. In this case, integrity is maintained as required by the standard for testing armor vests [42].

Fiber breaking, depicted in Figure 16, is a general mechanism common to natural and synthetic fibers, including the aramid fibers in Kevlar™ [61]. In principle, fiber breaking is an alternative to its pullout. In other words, a matrix well-adhered fiber will break instead of pulled-out. This is the case of GOCF composites in which the graphene oxide coating, Figure 9b, is expected to improve the curaua fiber adhesion to the epoxy matrix. Therefore, no pullout occurs in the GOCF fibers that comparatively dissipates more energy by breaking. It is interesting to observe in Figure 16 the rupture of an intact as well as a fibrils split curaua fibers, both indicated by corresponding arrows.

Matrix rupture exemplified in Figure 18 by a flat epoxy broken surface (right side) around a well-adhered GOCF fiber (left side). This is a specific mechanism for brittle polymer composites that undergo extensive matrix rupture upon a ballistic impact. A significant amount of energy is dissipated but enough well-adhered fibers, like in the present case of 30 vol% of GOCF, is important to avoid loss of integrity as shown in Figure 13.

As a final remark, it is worth reminding that the combination of energy dissipation mechanisms guarantees to a 30 vol% curaua fiber (plain or graphene oxide coated) reinforced epoxy composite as MAS second layer, an acceptable ballistic performance, Table 2, similar to that of a Kevlar™ laminate with same thickness. This performance, given by the standard backface signature less than 44 mm [42], is slightly superior in the GOCF composites, Table 2 and Figure 12, owing to the better fiber/matrix adhesion provided by the GO coating, in some of the aforementioned mechanisms. On the other hand, this better adhesion supports the 30 vol% GOCF integrity, which is essential for MAS in armor vest.

## 4. Conclusions

According to the FTIR analysis, the GO caused changes in the characteristic bands of the CF fibers, suggesting that bonds were formed as well as the appearance of new bands characteristic of the molecular structure of the GO.The thermal degradation of the GOCF fibers was retarded by the action of the GO coating, causing an insulation which contributes to higher temperature resistance, in relation to the CF fibers.Pullout test of untreated curaua fiber (CF) and graphene oxide coated curaua fiber (GOCF) embedded in epoxy matrix revealed a substantial reduction in the GOCF critical length in association with a more than 50 percent higher interfacial shear strength. This behavior is also superior to those of other material fibers.Epoxy composite plates reinforced with 30 vol% of either CF or GOCF, applied as 10 mm thick second layer in a front ceramic multilayered armor system, display a ballistic performance against the threat of 7.62 mm projectile within the backface signature (indentation < 44 mm) required by the standard.This ballistic performance comparable to that of the same thickness Kevlar™ laminate as MAS second layer, was for the first time interpreted as been related to a combination of the following impact energy mechanisms: (i) capture of fragments; (ii) fibrils separation; (iii) fiber pullout; (iv) composite delamination; (v) fiber breaking; and (vi) matrix rupture.The better adherence of GOCF to the epoxy matrix reduces, comparatively, the amount of absorbed energy by mechanisms (ii), (iii), (iv), and (vi). This results in slightly higher ballistic backface signature but a better integrity for the 30 vol% GOCF composites, which is a necessary condition for armor vest using MAS. The plain CF ballistic performance is similar to other natural fibers.It is also ruled out the need of a ductile metal sheet, usually applied as MAS third layer, since the 10 mm thick GOCF composite is enough for the required standard performance.

## Figures and Tables

**Figure 1 polymers-11-01356-f001:**
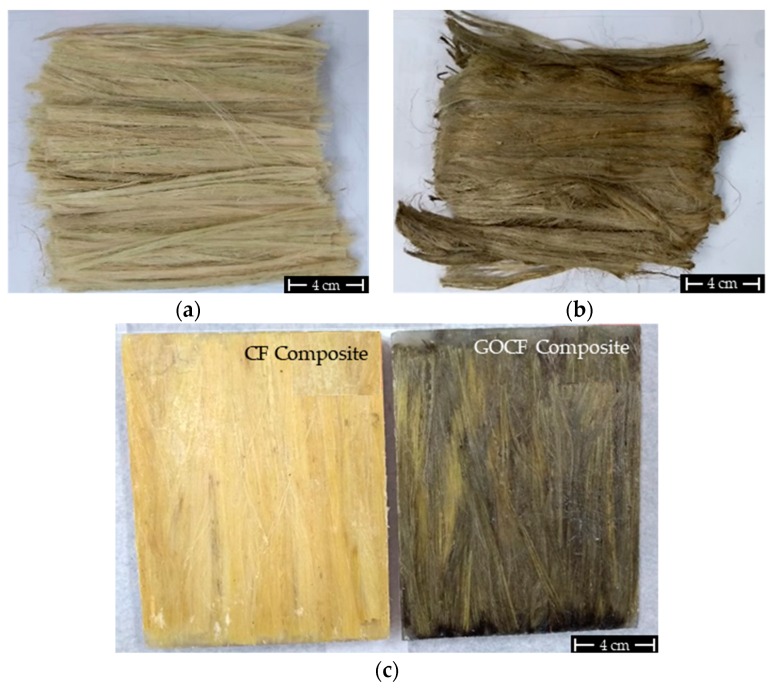
General macroscopic aspect of curaua fibers: (**a**) curaua fibers (CF); (**b**) graphene oxide coated fibers (GOCF); (**c**) their 30 vol% epoxy composites.

**Figure 2 polymers-11-01356-f002:**
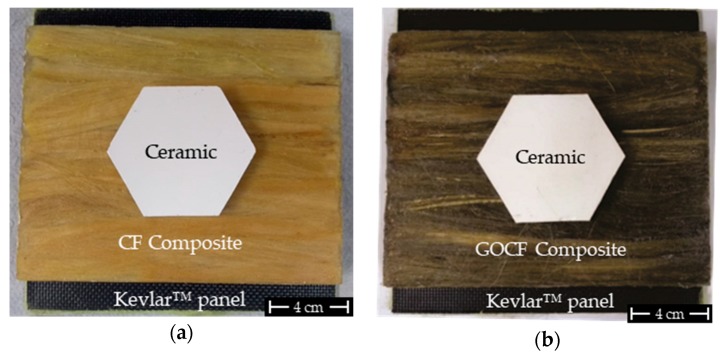
Multilayer armor system (MAS) mounted: (**a**) MAS with CF composite and (**b**) MAS with GOCF composite.

**Figure 3 polymers-11-01356-f003:**
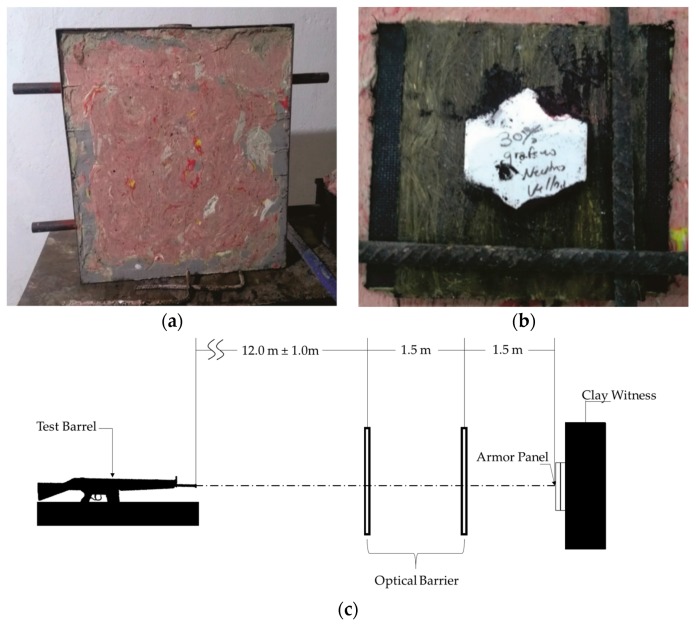
System used for ballistic tests: (**a**) Shooting support frame filled with clay witness; (**b**) MAS target ahead of the clay witness; (**c**) scheme of the system used for ballistic tests [42].

**Figure 4 polymers-11-01356-f004:**
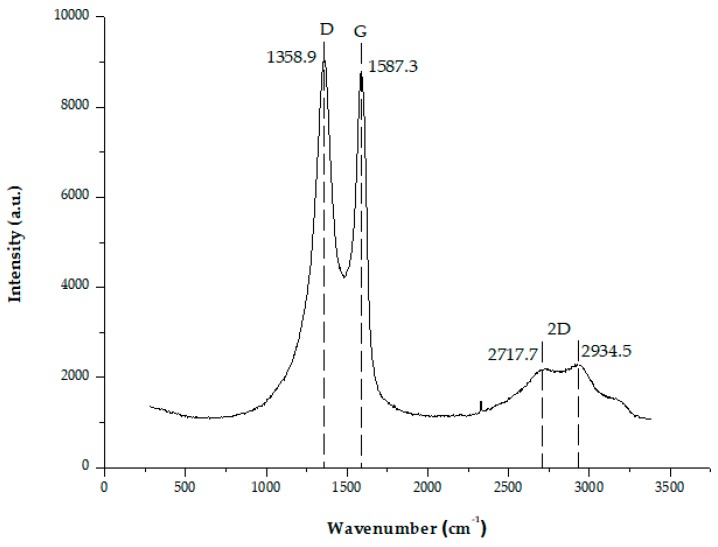
Raman spectra of GO colloid solution.

**Figure 5 polymers-11-01356-f005:**
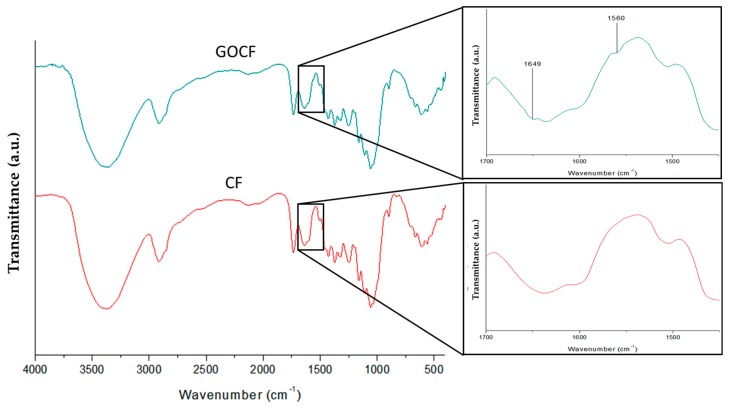
FTIR spectrum of CF and GOCF fibers.

**Figure 6 polymers-11-01356-f006:**
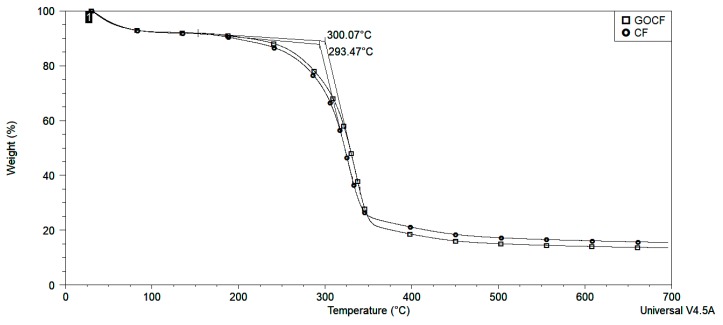
Thermogravimetry analysis (TGA) curves of CF and GOCF fibers.

**Figure 7 polymers-11-01356-f007:**
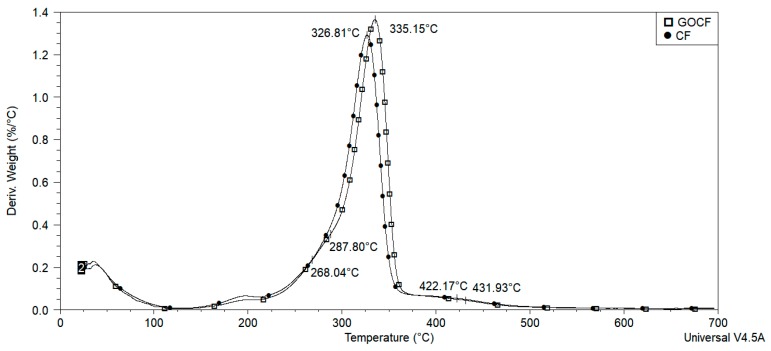
DTA curves of CF and GOCF fibers.

**Figure 8 polymers-11-01356-f008:**
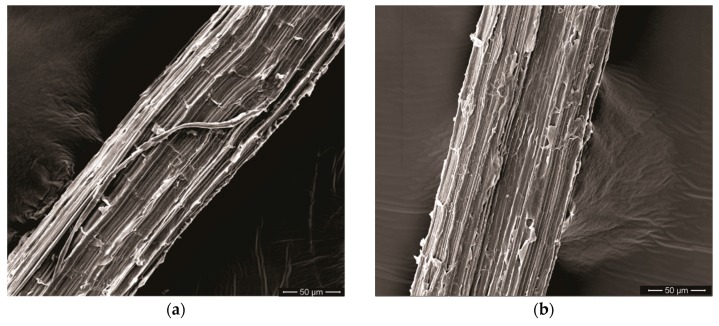
Scanning electron microscopy (SEM) micrographs of both investigated curaua fibers: (**a**) non-coated CF; (**b**) GOCF.

**Figure 9 polymers-11-01356-f009:**
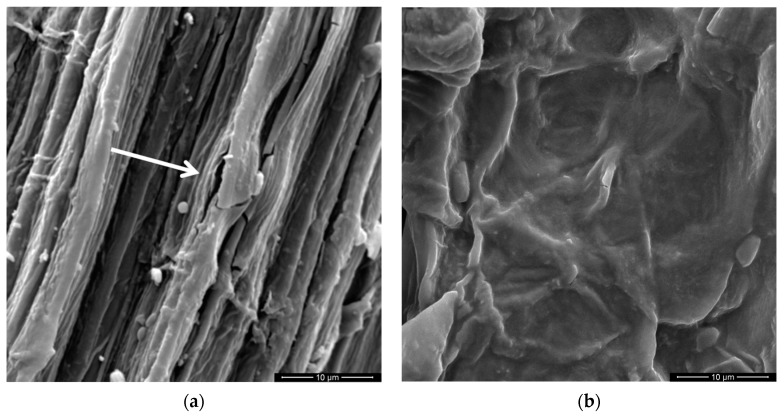
SEM surface images of both fibers: (**a**) CF; (**b**) GOCF.

**Figure 10 polymers-11-01356-f010:**
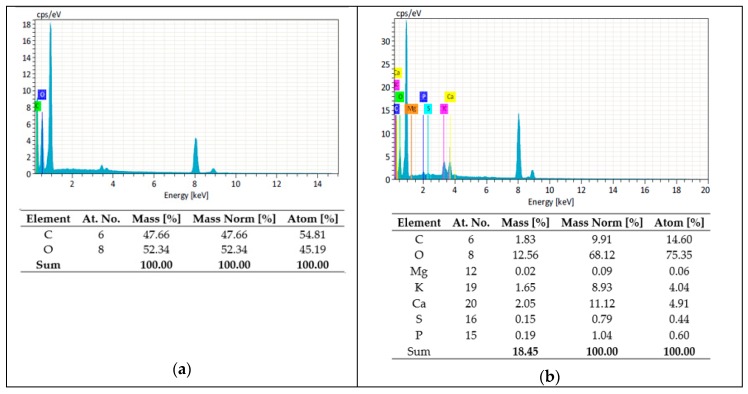
EDS pattern of CF and GOCF fibers: (**a**) CF; (**b**) GOCF.

**Figure 11 polymers-11-01356-f011:**
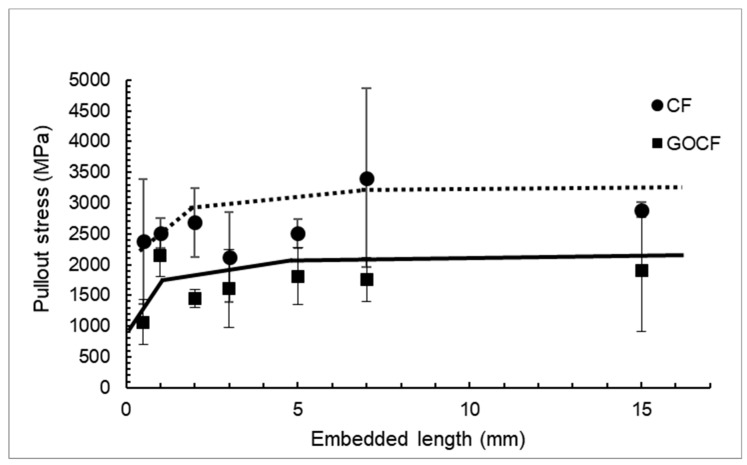
Pullout stress of both curaua fibers, CF and GOCF, versus epoxy embedded length curves.

**Figure 12 polymers-11-01356-f012:**
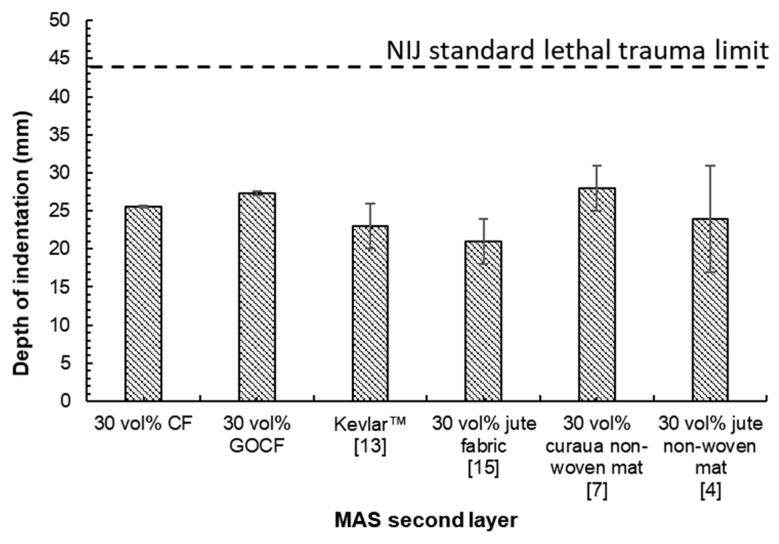
Depth indentation in clay witness of the reinforced composites with 30 vol%.

**Figure 13 polymers-11-01356-f013:**
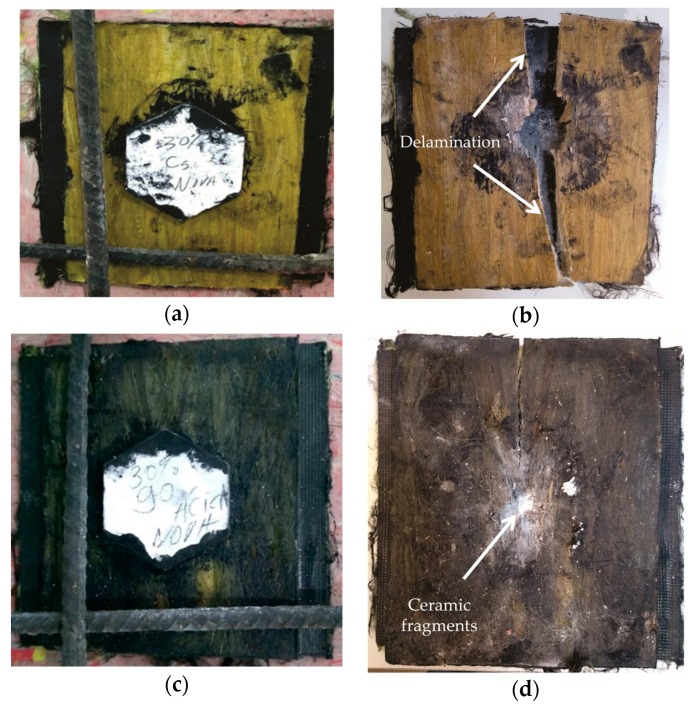
View of MAS target before (**a**,**c**) and after (**b**,**d**) the ballistic test: with second layer of (**a**,**b**) 30 vol% CF; (**c**,**d**) 30 vol% GOCF.

**Figure 14 polymers-11-01356-f014:**
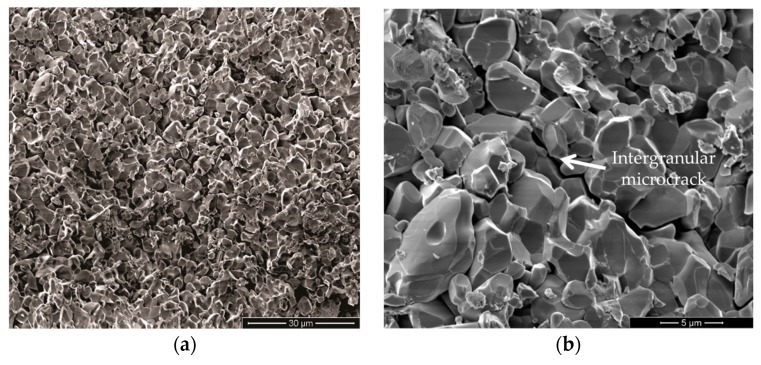
Surface of fracture of the ceramic tablets: (**a**) 3000×; (**b**) 10,000×.

**Figure 15 polymers-11-01356-f015:**
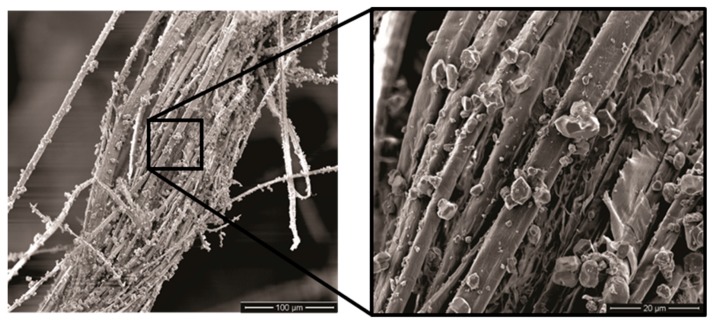
Curaua fiber covered with ceramic fragments.

**Figure 16 polymers-11-01356-f016:**
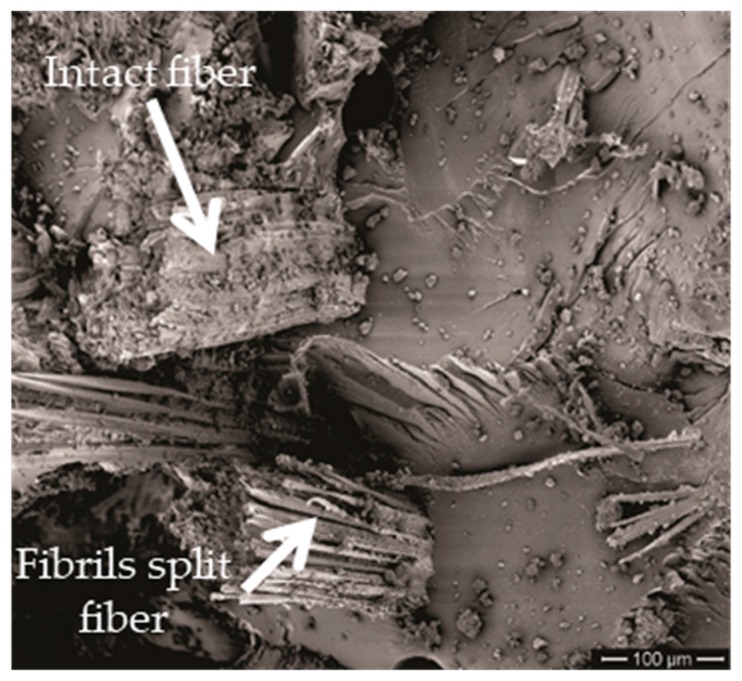
Fiber breaking of the GOCF composite fracture surfaces.

**Figure 17 polymers-11-01356-f017:**
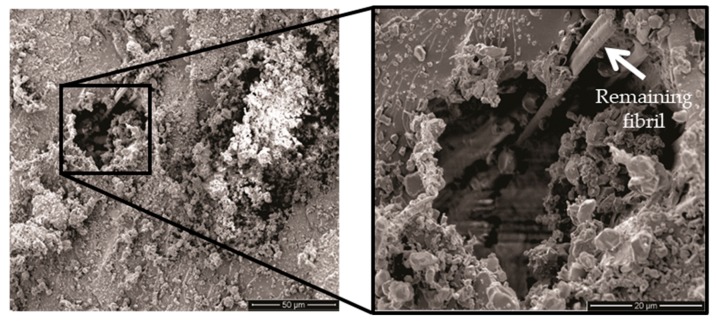
Fiber pullout of the CF composites.

**Figure 18 polymers-11-01356-f018:**
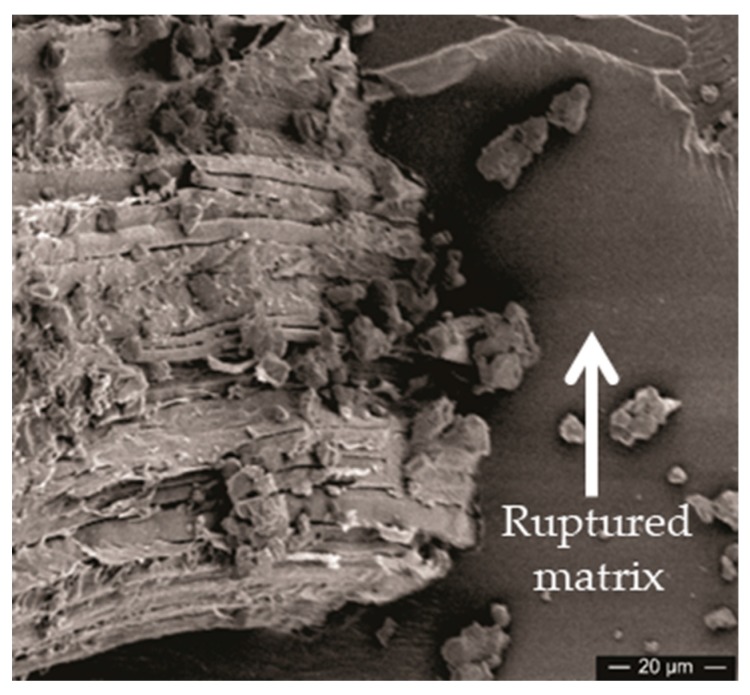
Matrix rupture of the GOCF Composite.

**Table 1 polymers-11-01356-t001:** Mechanical properties of CF and GOCF.

Condition	$\sigma_{f}\;(\mathrm{MPa})$	*ε* (%)	E (GPa)
CF	3153 ± 970	13.48 ± 5.45	25.7 ± 11.3
GOCF	1834 ± 673	8.82 ± 3.10	38.0 ± 10.0

**Table 2 polymers-11-01356-t002:** Depth of indentation of MAS with natural fibers composites and same thickness Kevlar™ for comparison.

MAS Second Layer	Depth of Indentation (mm)	Reference
30 vol% curaua fiber/epoxy composite	25.6± 0.2	PW
30 vol% curaua fiber coated with GO/epoxy composite	27.4 ± 0.3	PW
Kevlar™	23 ± 3	[13]
30 vol% jute fabric/epoxy composite	21 ± 3	[15]
30% curaua non-woven mat/epoxy composite	28 ± 3	[7]
30 vol% jute non-woven mat/polyester composite	24 ± 7	[4]

PW—Present work.

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
