# Peer review of "Effect of Graphene Oxide Coating on Natural Fiber Composite for Multilayered Ballistic Armor"

_polymers, 2019, doi:10.3390/polym11081356_

Round 1
Reviewer 1 Report
Manuscript: Effect of Graphene Oxide Coating on Natural fiber 2 Composite for Multilayered Ballistic Armor
Manuscript ID: Polymers-530261
Manuscript Presents good research work but required minor revision before consideration to publication. Some minor comments are as follows.
(1) Authors need to incorporate some interesting data in the abstract part of the manuscript.
(2) Authors needs to incorporate recent reference related to preparation and application of sustainable natural fibers and derived materials (specially in introduction part). For example;
(a) Angew. Chem. Int. Ed. 2005, 44, 3358 – 3393 (b) materialstoday Volume, 21, Issue 7, September 2018, Pages 720-748 (c) Biomacromolecules 18 (8), 2333-2342 (d) ACS Sustainable Chemistry & Engineering 6 (3), 3279-3290 (e) Industrial & Engineering Chemistry Research 56 (46), 13885-13893 (g) ACS Applied Nano Materials 1 (8), 3969-3980 (h)ACS Sustainable Chem. Eng., 2019, 7 (6), pp 6140–6151 (i) Chem. Mater. 2017, 29, 5426−5446 (k) J. Renew. Mater., Vol. 4, No. 5, October 2016 (l) Nano Energy 35 (2017) 299–320
(3) Authors need to provide FT-IR data for non-coated fibers (CF) and graphene oxide coated fibers (GOCF) samples.
(4) Authors need incorporate TGA-DTA profile with proper explanations for non-coated fibers (CF) and graphene oxide coated fibers (GOCF) samples.
(5) Author need to correct typos throughout the manuscript (Figure word is written in different formats).
(6) Figure 6, authors need to provide EDX pattern to understand the chemical composition for CF, GOCF samples.
(7) Figure 7, Embedded Length must be written as Embedded length.
(8) Author need to label important peaks in Figure 4.
(9) It will be good if authors will compare their results with similarly reported research in the conclusion part of the manuscript.
(10) What kind of interaction observed between graphene oxide and natural fibers? Suitable explanation required.
Author Response
Polymers 554260
The authors would like to thank the Reviewers for the valuable comments and suggestions that contribute to improve our manuscript.
Amendments were provided accordingly, and all modifications were marked as Track changes in the revised version of the manuscript. Responses to each comment, point by point, are given below.
Reviewer 1
General Comment: Manuscript Presents good research work but required minor revision before consideration to publication. Some minor comments are as follows.
Response: The authors acknowledge the Reviewers for the favorable comment regarding our research work.
Comment 1: Authors need to incorporate some interesting data in the abstract part of the manuscript.
Response: Important data related to the results and discussion are now incorporated to the abstract.
Comment 2: Authors needs to incorporate recent reference related to preparation and application of sustainable natural fibers and derived materials (specially in introduction part). For example: (a) Angew. Chem. Int. Ed. 2005, 44, 3358 – 3393 (b) materialstoday Volume, 21, Issue 7, September 2018, Pages 720-748 (c) Biomacromolecules 18 (8), 2333-2342 (d) ACS Sustainable Chemistry & Engineering 6 (3), 3279-3290 (e) Industrial & Engineering Chemistry Research 56 (46), 13885-13893 (g) ACS Applied Nano Materials 1 (8), 3969-3980 (h)ACS Sustainable Chem. Eng., 2019, 7 (6), pp 6140–6151 (i) Chem. Mater. 2017, 29, 5426−5446 (k) J. Renew. Mater., Vol. 4, No. 5, October 2016 (l) Nano Energy 35 (2017) 299– 320.
Response: Some of the suggested relevant references are now quoted in the Introduction.
Comment 3: Authors need to provide FT-IR data for non-coated fibers (CF) and graphene oxide coated fibers (GOCF) samples.
Response: FTIR for CF and GOCF sample are now presented in the revised version.
Comment 4: Authors need incorporate TGA-DTA profile with proper explanations for non-coated fibers (CF) and graphene oxide coated fibers (GOCF) samples.
Response: TGA – DTA for CF and GOCF are now presented with proper explanation.
Comment 5: Author need to correct typos throughout the manuscript (Figure word is written in different formats).
Response: A thorough revision was performed to correct types.
Comment 6: Figure 6, authors need to provide EDX pattern to understand the chemical composition for CF, GOCF samples.
Response: EDS pattern is now provided for the chemical composition of CF and GOCF
Comment 7: Figure 7, Embedded Length must be written as Embedded length.
Response: Complied
Comment 8: Author need to label important peaks in Figure 4.
Response: The important peaks in Fig 4 are now labeled
Comment 9: It will be good if authors will compare their results with similarly reported research in the conclusion part of the manuscript.
Response: Results with similarly reported research works are now compared in the conclusions.
Comment 10: What kind of interaction observed between graphene oxide and natural fibers? Suitable explanation required.
Response: A proper explanation is now provided for the possible interaction between the graphene oxide and natural fibers, particularly our curaua fiber in manuscript.
Reviewer 2 Report
The GO has been used to reinforce the interfacial stress transfer for many types of fibers. Here, GO coating was used for Curaua fibers, and the reinforced interface adhesion between fibers and epoxy results to better ballistic resistance. Overall, this manuscript has done systematic studies on the interface properties and ballistic resistance of curaua fiber/GO/epoxy composites, I suggest possible publication after proper revisions.
1. For caraua fiber/GO composite, what is the GO percentage in the composite? What is the optimal percentage? How is the uniformity of GO coating?
2. Line 149: The decima places should be consistent.
3. Figure 6: I don’t feel that the surface of GOCF is smoother than CF. The meaning of the arrows should be noted in the caption. Line 154: GOFC->GOCF.
4. Table 1: It is clear that the modulus of GOCF is higher than CF, while its strength is lower even after considering standard deviation. I don’t think that ‘these 160 basic tensile properties are practically similar for CF and GOCF.’
5. Figure 7: Why the pullout stress of CF is higher than GOCF at the same embedded length? If it is the case, then the interfacial strength of CF would be higher?
6. Figure 11: An EDS measurement is recommended to identify the components of these fragments.
Author Response
Polymers 554260
The authors would like to thank the Reviewers for the valuable comments and suggestions that contribute to improve our manuscript.
Amendments were provided accordingly, and all modifications were marked as Track changes in the revised version of the manuscript. Responses to each comment, point by point, are given below.
Reviewer 2
General comment: The GO has been used to reinforce the interfacial stress transfer for many types of fibers. Here, GO coating was used for Curaua fibers, and the reinforced interface adhesion between fibers and epoxy results to better ballistic resistance. Overall, this manuscript has done systematic studies on the interface properties and ballistic resistance of curaua fiber/GO/epoxy composites, I suggest possible publication after proper revisions.
Response: The authors thank the Reviewer for his words of appreciation regarding our manuscript.
Comment 2: L149 the decima places should be consistent.
Response: Complied
Comment 3: Figure 6: I don’t feel that the surface of GOCF is smoother than CF. The meaning of the arrows should be noted in the caption. Line 154: GOFC->GOCF.
Response: The reviewers is right; Fig. 6 does not give the visual impression of smother GOCF surface. This has been changed and better explanation is given including the meaning of the arrows. L154 is now corrected.
Comment 4: Table 1: It is clear that the modulus of GOCF is higher than CF, while its strength is lower even after considering standard deviation. I don’t think that ‘these 160 basic tensile properties are practically similar for CF and GOCF.’
Response: We fully understand and the Reviewer questioning the similarity of the tensile properties for CF and GOCF. A new interpretation is now provided to clarify our point of view.
Comment 5: Figure 7: Why the pullout stress of CF is higher than GOCF at the same embedded length? If it is the case, then the interfacial strength of CF would be higher?
Response: The Reviewer raised an important point regarding the pullout results. This is a relevant consequence for the interfacial shear strength, which is better discussed in the revised version.
Comment 6: Figure 11: An EDS measurement is recommended to identify the components of these fragments.
Response: The Reviewer raised an important point regarding the fragments that remains in the composite fracture after the ballistic impact. However, this topic has already been largely explained in the literature, and there are no novelties about this. The ceramic tile is made by Aluminum oxide (Al2O3) with Niobium oxide (Nb2O5) and the projectile is made by steel, therefore, the fragments are from both, ceramic and projectile. They stuck on the surface of the fiber by Van Der Waals bond.
Reviewer 3 Report
The article has both original content and a "review" of previous works. The results vs. kevlar are not significantly different (depth of indentation of 27.4 mm for GOCG and 23+3 mm for kevlar). However, the fact that the GOCF composite constitutes an alternative to Kevlar is already an achievement that can be positively valued.
The high recycling of contents from previous articles in the current manuscript reduces its originality (and the interest of this contribution to the reader), but it must be recognized that they have been integrated in a competent manner and it is tolerable. However, it seems necessary to advise the authors not to continue tempting devils and, in the case of this publication, it is strongly recommended to remove photographic material from previous contributions and, in particular, to redo Figure 3c. Apart from that, the non-correspondence between figures 6a and 6b renders the arrows on 6b ineffective (even more because they are not accompanied by an explanation).
Author Response
Polymers 554260
The authors would like to thank the Reviewers for the valuable comments and suggestions that contribute to improve our manuscript.
Amendments were provided accordingly, and all modifications were marked as Track changes in the revised version of the manuscript. Responses to each comment, point by point, are given below.
Reviewer 3
Comment 1: The article has both original content and a "review" of previous works. The results vs. Kevlar are not significantly different (depth of indentation of 27.4 mm for GOCG and 23+3 mm for Kevlar). However, the fact that the GOCF composite constitutes an alternative to Kevlar is already an achievement that can be positively valued.
Response: We thank the general positive evaluation of our article expressed by the Reviewer
Comment 2: The high recycling of contents from previous articles in the current manuscript reduces its originality (and the interest of this contribution to the reader), but it must be recognized that they have been integrated in a competent manner and it is tolerable. However, it seems necessary to advise the authors not to continue tempting devils and, in the case of this publication, it is strongly recommended to remove photographic material from previous contributions and, in particular, to redo Figure 3c. Apart from that, the non-correspondence between figures 6a and 6b renders the arrows on 6b ineffective (even more because they are not accompanied by an explanation).
Response: We fully agree with this relevant consideration by the Reviewer and the two recommendations were implemented.
- Photos from our previous works were removed or redone as is the case of Fig. 3c.
A better explanation given to the correspondence between Fig. 6a and 6b, justifying the arrows in Fig. 6b.